

# Hybrid Hydrological Modeling for Large Alpine Basins: A Distributed Approach

Bu Li[1], Ting Sun[2], Fuqiang Tian[1], Mahmut Tudaji[1], Li Qin[3], and Guangheng Ni[1]

[1]State Key Laboratory of Hydro-science and Engineering, Department of Hydraulic Engineering, Tsinghua University, Beijing 100084, China
[2]Institute for Risk and Disaster Reduction, University College London, London WC1E 6BT, UK
[3]Gansu Academy for Water Conservancy, Lanzhou 730030, China

**Correspondence:** Guangheng Ni (ghni@tsinghua.edu.cn)

**Abstract.** Large alpine basins provide abundant water resources crucial for hydropower generation, irrigation, and daily life. It is thus crucial to develop high-performance hydrological models for water resources management in large alpine basins. Recently, hybrid hydrological models have come to the forefront, synergizing the exceptional learning capacity of deep learning with the interpretability and physical consistency of process-based models. These models exhibit considerable promise in

achieving precision in hydrological simulations. However, a notable limitation of existing hybrid models lies in their failure to incorporate spatial information within the basin and describe alpine hydrological processes, which restricts their applicability in hydrological modeling in large alpine basins. To address this issue, we develop a set of hybrid distributed hydrological models by employing a distributed process-based model as the backbone, and utilizing embedded neural networks (ENNs) to parameterize and replace different internal modules. The proposed models are tested on three large alpine basins on the

Tibetan Plateau. Results are compared to those obtained from hybrid lumped models, state-of-the-art distributed hydrological model, and purely deep learning models. A climate perturbation method is further used to test the applicability of the hybrid models to analyze the hydrological sensitivities to climate change in large alpine basins. Results indicate that proposed hybrid hydrological models can perform well in predicting runoff processes and simulating runoff component contributions in large alpine basins. The optimal hybrid model with Nash-Sutcliffe efficiency coefficients ($NSEs$) higher than 0.87 shows compara-

ble performance to state-of-the-art DL models. The hybrid distributed model also exhibits remarkable capability in simulating hydrological processes at ungauged sites within the basin, markedly surpassing traditional distributed models. Besides, the results also show reasonable patterns in the analysis of the hydrological sensitivities to climate change. Runoff exhibits an amplification effect in response to precipitation changes, with a 10% precipitation change resulting in a 15–20% runoff change in large alpine basins. An increase in temperature enhances evaporation capacity and changes the redistribution of rainfall and

snowfall and the timing of snowmelt. It further leads to a decrease in the total runoff, the contributions of snowmelt runoff, and the intra-annual variability of runoff. Overall, this study provides a high-performance tool enriched with explicit hydrological knowledge for hydrological prediction and improves our understanding about the hydrological sensitivities to climate change in large alpine basins.



## 1 Introduction

The large alpine basins almost offer abundant water resources crucial for hydropower generation, irrigation, and daily life (Cui et al., 2023; Huss et al., 2017; Viviroli et al., 2011). Developing accurate hydrological models in large alpine basins is of paramount importance in effectively addressing challenges posed by floods and droughts, as well as in improving water usage efficiency (Blöschl et al., 2019).

Process-based hydrological models, such as EXP-Hydro (Patil and Stieglitz, 2014), CRHM (DeBeer and Pomeroy, 2017),

and THREW (Nan et al., 2021), are widely used approaches for hydrological simulation in large alpine basins. These models use established physical laws or empirical relationships to describe physical processes and are grounded in well-defined physical mechanisms. They can be used to understand the entire hydrological systems including all internal processes and assess the response of hydrological systems to changes in the driving forces or properties, which are essential to advance scientific understanding and can demonstrate that the simulated results are regarded as highly reliable (Cui et al., 2023; Li et al., 2021).

However, the performance of these models is constrained by several factors, including an incomplete understanding of alpine hydrological processes, errors in the model structure, and uncertainties in parameterization (Kuppel et al., 2018; Beven, 2006). These deficiencies also give rise to equifinality, making it challenging to accurately represent hydrological processes. This diminishes the credibility of process-based models in the context of climate change assessment.

Deep learning (DL) hydrological models are distinguished by their remarkable data mining capabilities, operating indepen-

dently of hydrological knowledge. They have showcased exceptional model performance across diverse hydrological domains, encompassing runoff (Kratzert et al., 2018; Lees et al., 2021; Liu et al., 2021), snow water equivalent (Duan and Ullrich, 2021), and groundwater level (Solgi et al., 2021; Nourani et al., 2022). Most of these studies disregard the effect of spatial information from meteorological data on hydrological modeling. Li et al. (2023a) introduced an innovative spatiotemporal DL hydrological model, demonstrating that integrating spatial information can significantly improve the performance of DL mod-

els in hydrological modeling. Nonetheless, despite their remarkable capabilities, DL hydrological models still face scrutiny within the hydrological modeling community, primarily due to their "black-box" nature. Furthermore, DL models rely on the assumption that the dataset's distribution during the prediction period remains consistent with that of the training period. This assumption cannot be met when using DL models to assess the effects of climate change on hydrological modeling (Nearing et al., 2021; Zhong et al., 2023).

Hybrid hydrological models that combine process-based and DL approaches are anticipated to harness their respective strengths to achieve both impressive performance and a well-defined understanding of hydrological processes (Tsai et al., 2021; Shen et al., 2023). Previous studies have introduced various hybrid model configurations and demonstrated satisfactory outcomes (Feigl et al., 2022; Frame et al., 2021; Kashinath et al., 2021; Quilty et al., 2022; Bhasme et al., 2022; Kumanlioglu and Fistikoglu, 2019; Xie et al., 2021; Lu et al., 2021), while the underlying concept in many of these hybrid models remains

centered on either pure DL models or process-based models. For instance, Frame et al. (2021) utilized LSTM models as postprocessors for the United States National Water Model, highlighting that integrating DL models can improve performance by rectifying errors in the outcomes of process-based models. Xie et al. (2021) introduced a physically-guided LSTM model





by incorporating synthetic samples during model training to capture underlying physical mechanisms. Recently, some studies attempted to implement differentiable models to facilitate a bidirectional integration between process-based models and DL
models (Shen et al., 2023; Baydin et al., 2018; Höge et al., 2022). Feng et al. (2022) introduced hybrid hydrological models that integrated a lumped hydrological model HBV as the foundation and incorporated embedded neural networks (ENNs) to parameterize, enhance, or replace internal components without prior training. The proposed models demonstrated comparable performance to DL models and can output untrained physical variables. Our earlier work further developed hybrid models by employing ENNs to replace the internal modules of the lumped model EXP-Hydro, and systematically test the impact of
replacing different internal modules with ENNs (Li et al., 2023b). The findings suggest that substituting any internal component with ENNs can enhance model performance, but increasing the number of internal component replacements does not guarantee improved outcomes. Achieving optimal performance requires a delicate equilibrium between the quantity of ENNs and the process constraints inherent in the process-based model. However, Feng et al. (2022) and Li et al. (2023b) have predominantly employed lumped hydrological models as the foundational framework in hybrid models. They have not adequately accounted
for the spatial information of meteorological inputs and underlying surfaces within the basin, which limits their applicability in large basins. Additionally, the effectiveness of hybrid models in the Tibetan Plateau's large alpine basins, particularly in assessing hydrological sensitivities to climate change, is yet to be clearly established. Therefore, there is a need to evolve hybrid models from lumped to distributed to adequately capture the spatial information within the basin. Moreover, it is also essential to incorporate alpine hydrological processes in hybrid models for adapting them to alpine basins and evaluate the
adaptability of these hybrid models in analyzing the hydrological sensitivities to climate change in large alpine basins.

Building upon our earlier work about hybrid lumped models (Li et al., 2023b), this study aims to propose hybrid distributed models that employ a distributed hydrological model as the backbone, and employ the ENNs to parameterize and replace different internal modules within the sub-basin scale. The proposed models are then comprehensively assessed across three large mountainous basins on the TP. A climate perturbation method is further used to analyze the hydrological sensitivities to
climate change in large alpine basins. The remainder of this paper is organized as follows: Sect. 2 outlines the proposed hybrid models, study area, and data, Sect. 3 shows the evaluation results of the proposed models, Sect. 4 provides details about the hydrological sensitivities to climate change, and we conclude in Sect. 5.

## 2  Methods and Materials

### 2.1  Model development

This study develops hybrid distributed hydrological models by integrating the distributed process-based model and embedded neural networks (ENNs; Figure 1). Specifically, proposed models use distributed EXP-Hydro model as the backbone, with ENNs parameterizing and replacing different internal modules. The differential programming framework is utilized to achieve a bidirectional integration between the process-based model and ENNs, enabling simultaneous parameter training of both entities.





### 2.1.1 The distributed EXP-Hydro model


In this study, the hybrid distributed models are built upon the foundation of the distributed EXP-Hydro model (Patil et al., 2014). The originally lumped EXP-Hydro model, proposed by Patil and Stieglitz (2014), treats each basin as a singular areal unit, disregarding the spatial information within the basin. The EXP-Hydro model encompasses a snow accumulation bucket and a basin bucket represented by snow storage ($S_0$) and basin water storage ($S_1$), respectively. Within the model, four processes

are represented: precipitation partition (rainfall $P_r$ or snowfall $P_s$), evapotranspiration ($ET$), snowmelt ($M$), and runoff ($Q$). Detailed equations refer to Appendix A1 and Patil and Stieglitz (2014). The distributed EXP-Hydro model was subsequently extended to incorporate the spatial heterogeneity within the basin (Patil et al., 2014). Initially, the study basin is divided into multiple sub-basins using a Digital Elevation Model (DEM). The EXP-Hydro model is run independently within each sub-basin, and the overall basin runoff is derived by summing the runoff outputs from all sub-basins (Equation A12). Patil and

Stieglitz (2015) and Patil et al. (2014) showcased the efficacy of the distributed EXP-Hydro model in hydrological modeling across 295 basins spanning the continental United States. Their studies indicated that this model outperforms the original EXP-Hydro model.

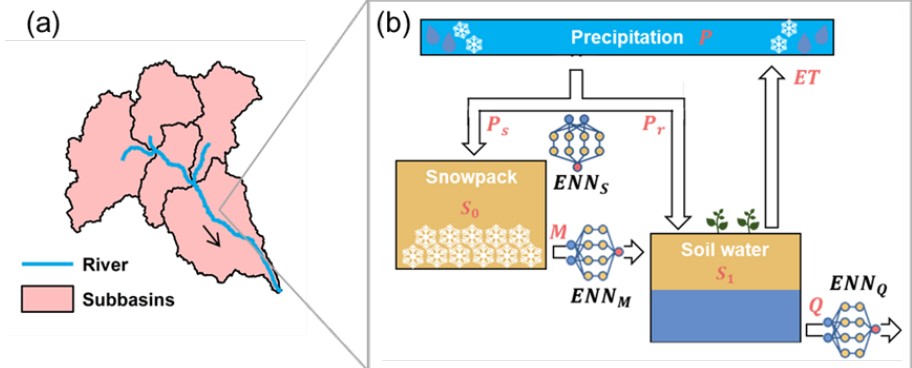

**Figure 1.** The schematic diagram of hybrid distributed models. (a) The basin is first divided into many sub-basins; (b) All meteorological and hydrological processes included in the EXP-Hydro model are calculated in each sub-basin. The precipitation partition, snowmelt, and runoff modules can be optionally replaced by embedded neural networks. Detailed formulations of these processes refer to the main text.

### 2.1.2 The hybrid distributed models

Using the distributed EXP-Hydro model as the backbone, the hybrid models integrate ENNs to parameterize and replace various

internal modules within the differential programming framework (Baydin et al., 2018). This configuration enables the model to comply with basic physical principles while enhancing its representational capability of the corresponding meteorological and hydrological modules, thus increasing the accuracy of hydrological simulations. ENNs utilize both static attributes (Table A1) and dynamic meteorological time series from each sub-basin as inputs. These inputs are employed to characterize the



disparities in physical mechanisms among sub-basins and to drive the precipitation-runoff processes. The hybrid models are realized via four steps:

1. Data pre-processing: DEM is employed to partition the study basin into multiple sub-basins, guided by a drainage area threshold (Grieve et al., 2016; Noël et al., 2014). The static attributes (Table A1) and daily meteorological time series for each sub-basin are derived by calculating the areal averages from the original dataset.

2. Distributed model development within the differential programming framework: All equations within the distributed models are formulated to be differentiable to ensure operating within the differential programming framework (Shen et al., 2023; Li et al., 2023b; Levine et al., 2016). This framework facilitates the computation of derivatives from model outputs to inputs and intermediate variables, thus enabling an "end-to-end" training approach. The hybrid model achieves simultaneous training of both the distributed hydrological models and ENNs. Only runoff data is employed as the training target, eliminating the need for observed data for ENN outputs. Furthermore, a physical recurrent neural network (P-RNN) is established to simulate hydrological dynamic processes and retain the memory of past basin storage sequences (Li et al., 2023b; Jiang et al., 2020).

3. ENNs parameterization and replacement: The calibration parameters of the distributed EXP-Hydro model are assumed to be the same in every sub-basin, while many of them related to sub-basin attributes should be different (Feng et al., 2022). To capture the spatial diversity of these calibration parameters at the sub-basin scale, we build an ENN to derive calibration parameters only using static attributes as inputs. Additionally, ENNs are employed to potentially substitute distinct internal modules of the distributed EXP-Hydro model, utilizing static attributes and corresponding dynamic time series as inputs. Specifically, three ENNs are designed for simulating runoff, precipitation partition, and snowmelt processes in this study.

4. Model training: Through the aforementioned steps, all parameters of the hybrid models, encompassing the distributed EXP-Hydro model and ENNs, can be jointly trained using observed runoff data as the training target. The Nash-Sutcliffe efficiency ($NSE$, (Nash and Sutcliffe, 1970)) is utilized as the loss function during training.

Our previous study has shown that the utilization of ENNs to substitute internal components of lumped hydrological models can elevate model performance in hydrological modeling (Li et al., 2023b). ENNs possess the flexibility to optionally replace any single or multiple internal modules of the distributed hydrological model. Similar to Li et al. (2023b), the ENN dedicated to precipitation partition employs precipitation and air temperature as inputs to compute the snowfall ratio. Rainfall is then determined by subtracting snowfall from the precipitation. The snowmelt ratio is determined through an ENN that takes air temperature as the input. The ENN related to the runoff process is developed using basin water storage, the combined value of rainfall and snowmelt, and air temperature as inputs. The inclusion of air temperature serves to depict the influence of soil freeze-thaw dynamics on the runoff process in alpine basins within the TP (Zhong et al., 2023). Apart from the dynamic driving time series, all the ENNs utilized for replacing internal components also incorporate static attributes as inputs, aiming to



differentiate disparities among various sub-basins. The detailed ENNs inputs refer to Appendix B. $ENN_Q$, $ENN_S$, and $ENN_M$ are utilized hereinafter to denote the ENN that replace runoff, precipitation partition, and snowmelt processes, respectively.

In this study, we develop and evaluate five hybrid distributed models denoted as $DM_\theta$, $DM_{\theta-Q}$, $DM_{\theta-Q-T}$, $DM_{\theta-QSM}$,

and $DM_{\theta-QSM-T}$ in this study. The $DM_\theta$ model solely employs the ENN for parameterizing calibration parameters across sub-basins. The $DM_{\theta-Q}$ and $DM_{\theta-Q-T}$ models go a step further by incorporating ENNs to replace the runoff process. Expanding upon this, the precipitation partition and snowmelt processes are substituted by corresponding ENNs in $DM_{\theta-QSM}$ and $DM_{\theta-QSM-T}$ models. Notably, the inputs for the $ENN_Q$ include air temperature in $DM_{\theta-Q-T}$ and $DM_{\theta-QSM-T}$ models, while $DM_{\theta-Q}$ and $DM_{\theta-QSM}$ models do not consider it.

### 2.1.3 Comparison models

We also compare our proposed models with the state-of-the-art distributed hydrological model THREW (Tsinghua Representative Elementary Watershed) and deep learning models LSTM and CNN-LSTM. The THREW model, originally proposed by Tian et al. (2006), operates by delineating the basin into representative elementary watersheds (REWs) through DEM calculation. Furthermore, each REW is subdivided into sub-zones, which serve as the fundamental units for hydrological modeling. The THREW model has demonstrated successful applications across diverse basins, including representative ones within the

Tibetan Plateau, Alps, and Tianshan (Cui et al., 2023; He et al., 2014). To establish a fair comparison of model performance between the THREW model and the proposed hybrid models, the THREW model in this study is subjected to the same spatial discretization utilized by the hybrid models. LSTM models have recently shown excellent capabilities in hydrological simulation all over the world (Lees et al., 2021; Li et al., 2023a; Kratzert et al., 2019; Hochreiter and Schmidhuber, 1997). To benchmark against our proposed hybrid models, we have sourced the LSTM and CNN-LSTM model results from Li et al.

(2023a). These models are renowned for their superior accuracy in existing deep learning research within the study basins. Furthermore, we also include the hybrid lumped hydrological models $EXP_Q$ and $EXP_{QSM}$, proposed by Li et al. (2023b), for comparative evaluation. Their backbone model is the lumped hydrological model EXP-Hydro. This allows us to assess the effect of spatial information on hydrological modeling within hybrid frameworks. Notably, the $EXP_Q$ and $EXP_{QSM}$ employ the same dynamic time series inputs of ENNs for module replacement as the $DM_{\theta-Q-T}$ and $DM_{\theta-QSM-T}$ models, respec-

tively. Besides, DM and EXP are utilized hereinafter to denote distributed and lumped EXP-Hydro models if not specified otherwise.

### 2.2 Study area and data

#### 2.2.1 Study area

The Tibetan Plateau (TP; Figure 2), acclaimed as the "Third Pole" and the "Water tower of Asia", stands as the world's

highest plateau. The TP provides a significant source of abundant water resources crucial for the sustenance of downstream communities. To evaluate the performance of proposed hybrid models on large alpine basins, this study focuses on the source regions of three major river basins: the Yellow River, the Yangtze River, and the Lancang River. These basins are recognized





as extensive mountainous regions within the TP (Figure 2). Each of these study basins spans an area exceeding 90,000 km², characterized by diverse topography with elevation fluctuations exceeding 3,000 m. Previous studies has shown that the glacier

process has a minimal impact on runoff modeling in the three study basins, and it is neglected in this study (Cui et al., 2023). Hereinafter, Yellow, Yangtze, and Lancang are used to denote the corresponding source regions in this study.

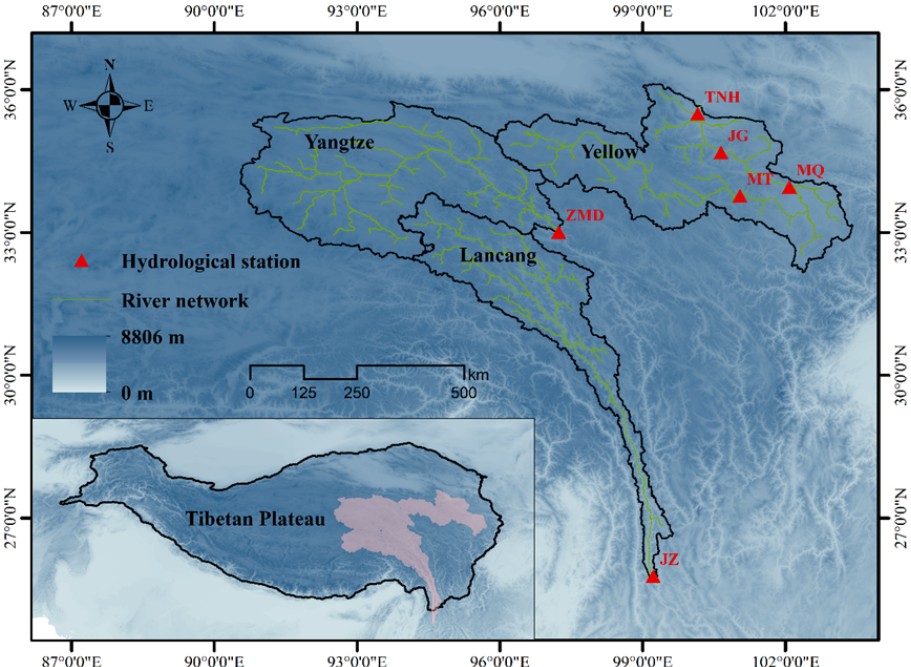

**Figure 2.** The terrain of the Tibetan Plateau and the location of the four study basins.

### 2.2.2 Data used

This study utilized the reanalysis and remote sensing datasets for input variables of hybrid models and the THREW model as follows:

1. Precipitation: China Meteorological Forcing Dataset (CMFD) with 0.1° spatial and 3h temporal resolution (Yang et al., 2010);

    2. Air temperature: The air temperature at 2m AGL (T2) from the fifth generation of ECMWF atmospheric reanalysis of the global climate (ERA5) reanalysis dataset with 0.1° spatial and 1 h temporal resolution (Hersbach et al., 2020);

    3. Potential evaporation: The potential evaporation from the ERA5 reanalysis dataset with 0.1° spatial and 1 h temporal

185        resolution (Hersbach et al., 2020);





4. DEM: Shuttle Radar Topography Mission (SRTM) with 90 m spatial resolution. The data set is provided by Geospatial Data Cloud site, Computer Network Information Center, Chinese Academy of Sciences. (http://www.gscloud.cn);

5. LAI: The MOD15A2H dataset from MODIS product with 500 m spatial and 8-day temporal resolution (Myneni et al., 2015);

190 6. NDVI: The MOD13A3 dataset from MODIS product with 1 km spatial and 1-month temporal resolution (Didan, 2015);

The daily observed runoff data at hydrological stations (Figure 2) is used for the model calibration/training and evaluation. The dataset is provided by local water agencies.

### 2.3 Experimental design

#### 2.3.1 Model evaluation schemes

195 We conduct two suites of experiments to comprehensively evaluate the performance of proposed hybrid distributed hydrological models in this study.

1. Model performance in trained sites: all proposed hybrid distributed models are developed, trained, and evaluated in three study basins. The comparison models are then utilized for a range of purposes: comparing the performance of the proposed models against state-of-the-art DL and distributed hydrological models, examining the effects of ENNs

200 parameterization and replacement on hydrological modeling, and appraising the impact of spatial information on model performance. Due to the limitation of observed runoff data, TNH in Yellow, ZMD in Yangtze, and JZ in Lancang are utilized as the evaluation stations in this experiment. For the Yellow and Yangtze, the training and evaluation periods are respectively designated as 1982–2007 and 2009–2014. In the case of the Lancang, these periods span 1988–2003 and 2005–2010.

205 2. Model performance in untrained sites within the basin: by capturing the spatial heterogeneity within the basin, hybrid distributed models provide the opportunity to predict hydrological processes at any untrained sites within the basin. To assess the proficiency of hybrid distributed models in ungauged sites within the basin, the MT, MQ, and JG stations, situated upstream of the TNH station in the Yellow (Figure 2), are simulated using Yellow (TNH) hydrological models in this section. The evaluation phase encompasses the years 2009 to 2014 for all hydrological stations.

210 #### 2.3.2 The climate perturbation method

This study uses the climate perturbation method to test the applicability of the hybrid models to analyze the hydrological sensitivities to climate change in three large alpine basins. Using precipitation and temperature data from the reanalysis dataset (Sect. 2.2.2) as the reference, the additional perturbation sequences are added to represent the potential climate changes. Perturbed precipitation sequences are extracted by multiplying the reference precipitation data from 80% to 120% with an

215 increment of 10% (Su et al., 2023). Perturbed temperature sequences are generated by adding from 0.5 to 2 °C with an




increment of 0.5 °C to the reference temperature input (Cui et al., 2023). The impact of increased temperature on the potential evapotranspiration is calculated by the regression between observed temperature and potential evapotranspiration in each sub-basin (Cui et al., 2023; Van Pelt et al., 2009; Xu et al., 2019). Total one reference, four perturbed temperature, and four perturbed precipitation sequences are conducted to assess the influence of precipitation and temperature change on hydrological

processes. The changes of other underlying surfaces are not considered in this study.

### 2.3.3 Evaluation metrics

Three common hydrological metrics – including NSE, modified NSE ($mNSE$; (Legates and McCabe Jr, 1999)), and the absolute value of peak flow bias ($PFAB$; (Yilmaz et al., 2008)) are employed to evaluate the model performance. They can be defined as follows:

$$NSE = 1 - \frac{\sum_{i=1}^{T}\left(Q_{obs,i} - Q_{sim,i}\right)^2}{\sum_{i=1}^{T}\left(Q_{obs,i} - \bar{Q_{obs}}\right)^2} \tag{1}$$

$$mNSE = 1 - \frac{\sum_{i=1}^{T}\left|Q_{obs,i} - Q_{sim,i}\right|}{\sum_{i=1}^{T}\left|Q_{obs,i} - \bar{Q_{obs}}\right|} \tag{2}$$

$$PFAB = 100 \times \left| \frac{\sum_{l=1}^{L}\left(Q_{sim:l} - Q_{obs:l}\right)}{\sum_{l=1}^{L} Q_{obs:l}} \right| \tag{3}$$

where $Q_{obs,i}$ and $Q_{sim,i}$ are the observed and simulated values, $T$ is the length of the evaluation period, and $\bar{Q_{obs}}$ is the averaged observed values. $Q_{sim:l}$ and $Q_{obs:l}$ are the observed and simulated runoff sorted in descending order, respectively. $L$

is the number of flow values which are in the top 2% of all flows. Both $NSE$ and $mNSE$ measure the overall fit-of-goodness of simulated and observed data, while $mNSE$ gives less weight to high values than NSE and thus focuses on the baseflow. A $NSE$ and $mNSE$ of 1 indicates the perfect fit and a $NSE$ of 0.55 is the threshold for good performance (Newman et al., 2015; Knoben et al., 2019). $PFAB$ emphasizes the performance for peak values and the value closer to zero indicates a smaller peak bias.

## 3 Model evaluation

### 3.1 Hybrid distributed model evaluation in trained sites

#### 3.1.1 The effect of ENNs on runoff modeling

In general, all hybrid distributed models exhibit notable performance, adeptly capturing the runoff peaks with appropriate magnitudes and timings across three study basins (Figure 3 and Table 1). Specifically, the comparisons with $DM_\theta$ and $DM$





models show that the $DM_\theta$ model slightly outperforms the $DM$ model in overall runoff modeling with an increase $NSE$ and $mNSE$ of 0.01-0.03 in all three basins. Additionally, lower $PFAB$ results imply that the $DM_\theta$ model contributes to an improved performance in peak runoff modeling. The incorporation of ENNs to represent spatial heterogeneity of calibration parameters is demonstrated to augment the ability of the distributed hydrological model to simulate both overall and peaking runoff processes.

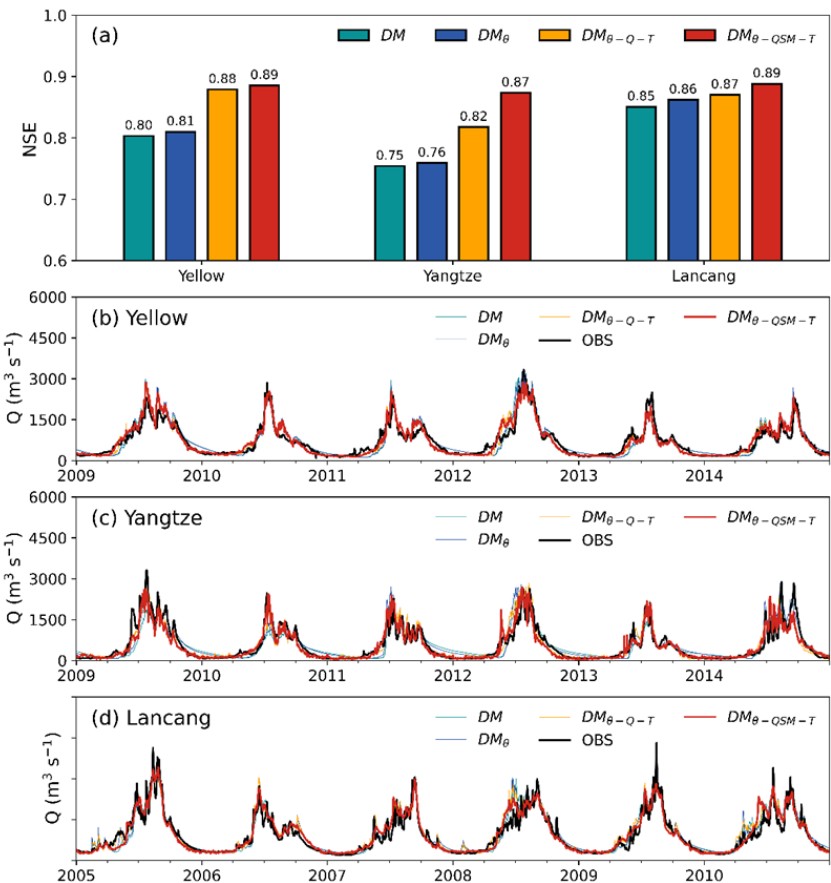

**Figure 3.** The comparison of simulated ($DM$, $DM_\theta$, $DM_{\theta-Q-T}$, and $DM_{\theta-QSM-T}$ models) and observed runoff processes in the evaluation period at the trained TNH, ZMD, and JZ station in Yellow, Yangtze, and Lancang, respectively.

The notably enhanced performance in $DM_{\theta-Q-T}$ and $DM_{\theta-QSM-T}$ models indicates that the inclusion of ENNs for replacing internal modules yields further improvements in model performance (Figure 3 and Table 1). First, the results between $DM_{\theta-Q-T}$ and $DM_\theta$ models show the substantial improvements in runoff modeling brought by the incorporation of $ENN_Q$. This enhancement is illustrated by a noteworthy increase in $NSE$ and $mNSE$ values, ranging from 0.06 to 0.09 in Yellow and Yangtze. Since the $DM_\theta$ model already exhibits commendable performance in Lancang, the advancements achieved by

the $DM_{\theta-Q-T}$ model are relatively marginal in comparison. $PFAB$ results suggest that the $ENN_Q$ does not lead to substan-





tial improvements in peak flow performance. Besides, evaluation findings for the $DM_{\theta-QSM-T}$ model show that replacing precipitation partition and snowmelt modules by ENNs can further improve the model performance with an increase $NSE$ of 0.01-0.05. It also does not translate into better peak runoff modeling as evidenced by comparable $PFAB$ scores across all three basins. ENNs employed for replacement in hybrid hydrological models have proven to be effective in enhancing the model

performance in runoff modeling. Among them, the $ENN_Q$ leads to the most substantial improvements in runoff prediction performance. The replacement of ENNs for snow-related processes ($ENN_S$ and $ENN_M$) results in comparatively minor enhancements. These findings align with our hydrological understanding as the runoff module directly generates runoff and thus plays a central role in runoff modeling. It thus contributes the most to the overall performance of runoff prediction. Conversely, the influence of snow-related processes on runoff modeling performance improvements is indirect and thus relatively modest

(Li et al., 2023b).

**Table 1.** The results of three hydrological metrics for different hybrid distributed models in three study basins.

| Basin | | $DM$ | $DM_\theta$ | $DM_{\theta-Q-T}$ | $DM_{\theta-QSM-T}$ | $DM_{\theta-Q}$ | $DM_{\theta-QSM}$ |
|---|---|---|---|---|---|---|---|
| TNH | $NSE$ | 0.8 | 0.81 | 0.88 | 0.89 | 0.87 | 0.88 |
| Yellow | $mNSE$ | 0.6 | 0.61 | 0.70 | 0.70 | 0.69 | 0.68 |
| | $PFAB$ | 6.68 | 5.18 | 6.48 | 4.57 | 9.46 | 10.84 |
| ZMD | $NSE$ | 0.75 | 0.76 | 0.82 | 0.87 | 0.83 | 0.84 |
| Yangtze | $mNSE$ | 0.56 | 0.59 | 0.66 | 0.73 | 0.66 | 0.7 |
| | $PFAB$ | 14.33 | 2.81 | 3.49 | 19.39 | 20.22 | 28.13 |
| JZ | $NSE$ | 0.85 | 0.86 | 0.87 | 0.89 | 0.87 | 0.87 |
| Lancang | $mNSE$ | 0.68 | 0.69 | 0.71 | 0.73 | 0.71 | 0.71 |
| | $PFAB$ | 10.41 | 9.19 | 7.97 | 8.64 | 8.16 | 9.1 |

The air temperature is employed as the additional input of the $ENN_Q$ to implicitly represent the soil freeze-thaw process in this study (Zhong et al., 2023; Gao et al., 2021). Results indicate that $DM_{\theta-Q-T}$ and $DM_{\theta-QSM-T}$ models exhibit slightly improved performance compared to the $DM_{\theta-Q}$ and $DM_{\theta-QSM}$ models, respectively. This enhancement is evident through higher $NSE$ and lower $PFAB$ values in Yellow and Lancang. While the $NSE$ values of these models are closely comparable

in the Yangtze basin, the $PFAB$ results underscore that incorporating air temperature can contribute to an enhanced capacity for predicting the runoff process. Moreover, the enhancement observed due to the inclusion of air temperature is notably more pronounced in Yellow and Yangtze compared to Lancang. This pattern aligns with expectations because Lancang features a smaller extent of permafrost regions, resulting in a lesser influence of the soil freeze-thaw process on runoff modeling in this region.

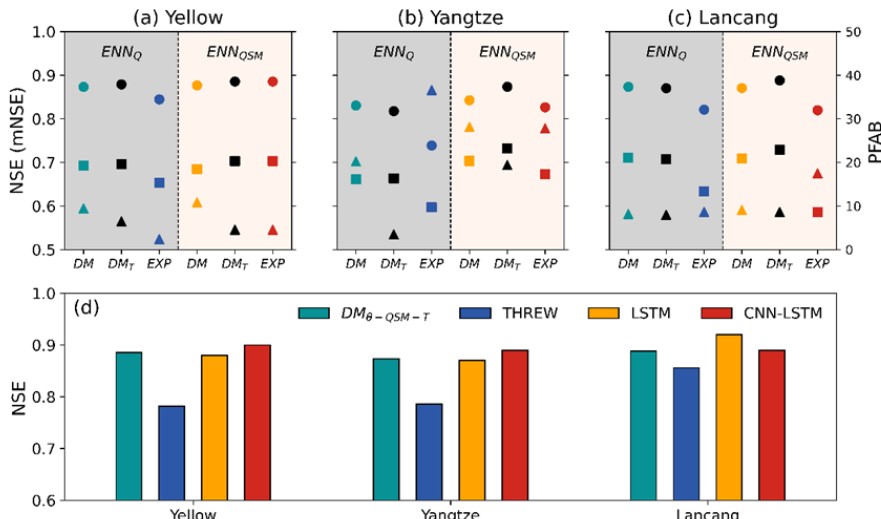

**Figure 4.** (a-c) The comparison of simulated and observed runoff processes in the evaluation period in Yellow, Yangtze, and Lancang, respectively. $DM_T$ and EXP are denoted to hybrid distributed and lumped models, while $DM$ represents the hybrid distributed models without inclusion of air temperature in $ENN_Q$. Circles, squares, and triangles refer to $NSE$, $mNSE$, and $PFAB$. (d) The model comparison with state-of-the-art models.

**Table 2.** The results of three hydrological metrics for different hybrid distributed and lumped models in three study basins.

| Basin | | $EXP_{\theta-Q}$ | $EXP_{\theta-QSM}$ | $DM_{\theta-Q-T}$ | $DM_{\theta-QSM-T}$ |
|---|---|---|---|---|---|
| TNH | $NSE$ | 0.84 | 0.85 | 0.88 | 0.89 |
| Yellow | $mNSE$ | 0.65 | 0.64 | 0.70 | 0.70 |
| | $PFAB$ | 2.42 | 8.65 | 6.48 | 4.57 |
| ZMD | $NSE$ | 0.74 | 0.83 | 0.82 | 0.84 |
| Yangtze | $mNSE$ | 0.6 | 0.67 | 0.66 | 0.7 |
| | $PFAB$ | 36.59 | 27.8 | 3.49 | 7.86 |
| JZ | $NSE$ | 0.82 | 0.82 | 0.87 | 0.89 |
| Lancang | $mNSE$ | 0.63 | 0.59 | 0.71 | 0.73 |
| | $PFAB$ | 8.66 | 17.48 | 7.97 | 8.64 |

### 3.1.2 The impact of spatial information on runoff modeling

Hybrid lumped models proposed by Li et al. (2023b) are similar with our proposed hybrid distributed models but did not consider the spatial heterogeneity. Hybrid lumped and distributed models are used to test the effect of spatial information on hydrological modeling. It is important to highlight that while the ENNs of the hybrid lumped models utilize the same dynamic





time series inputs as those of the distributed models, they do not include the static attributes of the basin. Results show that both
hybrid lumped models, $EXP_Q$ and $EXP_{QSM}$, exhibit strong performance in runoff modeling with $NSE$ more than 0.74 in
all three basins (Figure 4 and Table 2). It demonstrated the suitably of hybrid lumped models for hydrological modeling on
the TP. In comparison to $EXP_Q$ and $EXP_{QSM}$ models, the $DM_{\theta-Q-T}$ and $DM_{\theta-QSM-T}$ models show more impressive
performance in runoff modeling with the increase $NSE$ and $mNSE$ of 0.01-0.14 in three basins. $PFAB$ results affirm that
$DM_{\theta-Q-T}$ and $DM_{\theta-QSM-T}$ models excel in simulating peak flow processes, achieving $PFAB$ values of less than 10%
across all three basins. Consequently, the incorporation of spatial heterogeneity within the basin in hybrid models leads to
improved performance in both overall and peak runoff modeling. This finding is seamlessly consistent with our hydrological
comprehension and is also corroborated by related studies in the case of distributed process-based hydrological models and
DL hydrological models (Li et al., 2023a; Patil et al., 2014). In practice, we recommend the utilization of hybrid distributed
models for hydrological modeling, particularly in the context of large basins, to attain enhanced performance outcomes.

### 3.1.3  The comparison to the state-of-the-art models

We further use the optimal hybrid distributed model $DM_{\theta-QSM-T}$ to compare with state-of-the-art models: distributed hydro-
logical model THREW and DL models LSTM and CNN-LSTM (Li et al., 2023a). Results show that the $DM_{\theta-QSM-T}$ model
outperforms the THREW model by a substantial margin and holds comparable performance to the LSTM and CNN-LSTM
models (Figure 4). This reveals that our hybrid distributed model can effectively harness the advantages of both process-based
models and DL models. Specifically, it attains the high performance characteristic of DL models while adhering to the physical
mechanism constraints inherent in process-based models, creating a synergy not entirely realized in other models.

### 3.2  Hybrid distributed model evaluation in untrained sites within the basin

As proposed hybrid distributed models operate in a distributed manner, it is imperative to further investigate whether models
trained using the basin outlet point can effectively simulate hydrological processes in any untrained sites within the same
basin. In this study, runoff processes at three hydrological stations (JG, MQ, and MT), situated upstream of TNH in Yellow,
are simulated using our proposed distributed models trained by TNH data (Figure 2 and Figure 5).

Results reveal that all models trained on TNH data exhibit impressive predictive performance in simulating runoff processes
at JG and MQ stations, with $NSE$ values exceeding 0.71. The $DM_{\theta-QSM-T}$ model achieves an especially high $NSE$
of 0.84. However, the models show comparatively poorer performance in runoff modeling at the MT station, which is the
most upstream point in the basin (Figure 2 and Figure 5e). Hybrid distributed models with ENNs replacement, including
$DM_{\theta-Q-T}$ and $DM_{\theta-QSM-T}$ models, exhibit notably enhanced abilities in runoff modeling compared to $DM$ and $DM_{\theta}$
models, resulting in $NSE$ improvements ranging from 0.09 to 0.58. The $DM_{\theta-QSM-T}$ model demonstrates the strongest
performance in runoff modeling across all three stations, particularly in MT where its $NSE$ reaches 0.54, whereas the other
three models yield $NSE$ values lower than 0.22 (Figure 5). The findings show that the proposed hybrid distributed models
exhibit strong performance in hydrological modeling for untrained sites within the basin. It is also demonstrated that the
hydrological relationships established by ENNs are credible and robust.





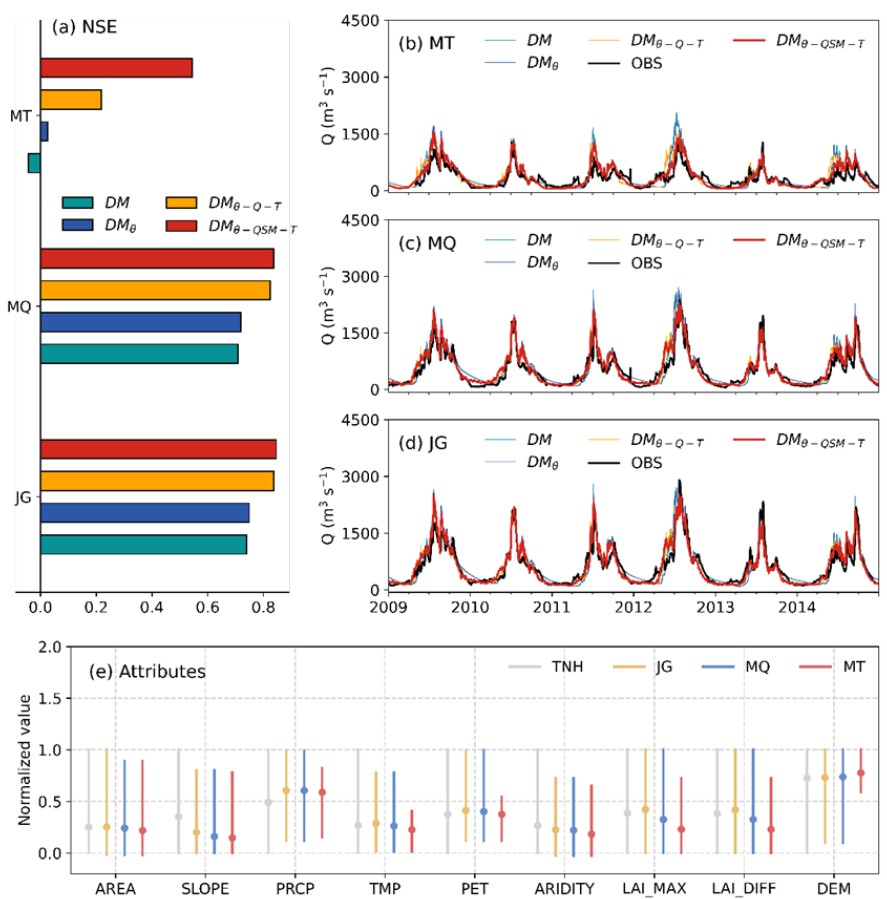

**Figure 5.** The $NSE$ results between simulated (different distributed hydrological models in TNH) and observed runoff processes at JG, MQ, and MT.

## 4 Hydrological sensitivities to climate change

Perturbed precipitation and air temperature dataset are input to trained $DM_{\theta-QSM-T}$ models to test the applicability of the hybrid models to analyze the hydrological sensitivities to climate change in three large alpine basins.

### 4.1 Sensitivities of runoff to perturbed precipitation

Figure 6 and Figure 7a-c depict the runoff sensitivities to various altered precipitation scenarios within three study basins. The findings suggest a consistent trend in the relationship between runoff and precipitation: runoff rises (decreases) as precipitation increases (decreases). Specifically, the annual runoff increases at rates of approximately 33.8, 18.1, and 44.9 mm/10% with the increase of precipitation within Yellow, Yangtze, and Lancang, respectively. The relative change in runoff surpasses that of precipitation in all three study basins: a 10% increase in precipitation leads to a 15% to 20% increase in runoff in all three



study basins. Besides, annual runoff exhibits greater sensitivity to increases in precipitation compared to decreases (Figure 7a-c). As an illustration, an increase of 20% in precipitation results in a substantial 40% increase in annual runoff, whereas a 20% decrease in precipitation leads to a notable 30% reduction in annual runoff in Yellow. It is indicated that runoff exhibits an amplification effect in response to precipitation changes due to the increase in the runoff coefficient with rising precipitation.

Figure 6a-c also illustrate that the inter-annual variation in runoff follows a pattern consistent with the annual runoff: there is a greater (lesser) variation in inter-annual runoff when there is an increase (decrease) in precipitation.

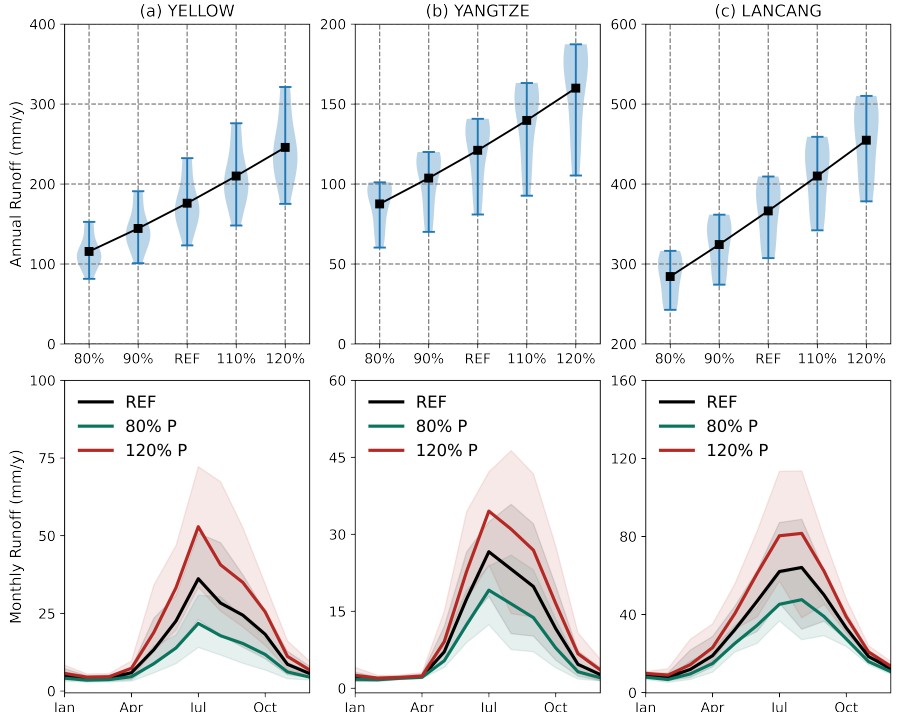

**Figure 6.** Runoff responses to altered precipitation in the Yellow, Yangtze, and Lancang basins (a-c for annual; d-f for monthly). The error bars in panels a-c and the shaded areas in panels d-f denote the range of simulated runoff

Moreover, the monthly runoff across all months shows a consistent response to perturbed precipitation, yet the extent of change varied among different months (Figure 6d-f and Figure 7a-c). Notably, the alterations during the wet seasons (June to October) are more pronounced compared to those in the dry seasons. This indicates that increased precipitation contributes

to a more concentrated distribution of runoff. Figure 6d-f also demonstrate that intra-annual runoff variation becomes more pronounced with higher levels of precipitation. These findings can be attributed to the fact that the augmented precipitation primarily occurs during the wet seasons, and the primary runoff components during these periods consist of direct rainfall runoff.

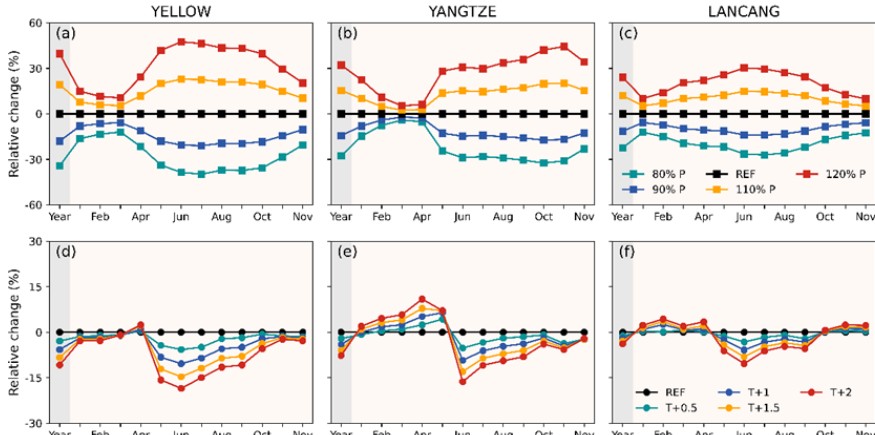

**Figure 7.** Relative change of annual (grey shadings) and monthly(yellow shadings) runoff response to the perturbed precipitation (a-c) and air temperature (d-f) in Yellow, Yangtze, and Lancang, respectively.

## 4.2 Sensitivities of runoff to perturbed temperature

The sensitivities of runoff to changing temperature follows a more intricate pattern: runoff tends to decrease as temperatures rise. This decrease is particularly pronounced during the flood season, while in the dry season, there is a slight increase in runoff (Figure 7d-f and Figure 8-9). This shift also leads to a reduction in the intra-annual variability of runoff. Taking the temperature increase of 2 °C as an example, the annual runoff in the three study basins decreases by less than 15%. When examining monthly runoff, the most significant increase occurs in April, while the most notable decrease is observed in June.

These phenomena can be explained by the fact that changes in temperature affect the evaporation capacity, the redistribution of rainfall and snowfall, and the timing of snowmelt. Higher temperature leads to increased evaporation capability, which results in more actual evaporation and less total runoff when precipitation remains constant. During the winter and spring, the increased rainfall and earlier snowmelt, along with higher actual evaporation, tend to balance each other, resulting in a minor increase or decrease in runoff. However, in the summer, reduced snowmelt and higher evaporation significantly reduce runoff.

To enhance the reliability of our model and validate our findings of hydrological sensitivities to climate change, we conducted an analysis of runoff component contributions in all three study basins across scenarios with varying temperature perturbations. It is essential to highlight that the glacier module has been excluded from this model due to structural limitations. Previous studies in the study basins have demonstrated that glaciers have a negligible impact on runoff (Cui et al., 2023; Su et al., 2022). As a result, this limitation does not significantly affect the accuracy of the simulation results. In the reference scenario,

rainfall runoff emerged as the primary component, contributing approximately 81.5%, 73.1%, and 84.0% to the total runoff in Yellow, Yangtze, and Lancang, respectively. Notably, these results align with findings from other studies (Cui et al., 2023; Su et al., 2022), underscoring that our hybrid model not only excels in simulating the runoff process but also accurately represents untrained hydrological processes. Furthermore, the contribution of snowfall runoff diminishes as the perturbed temperature



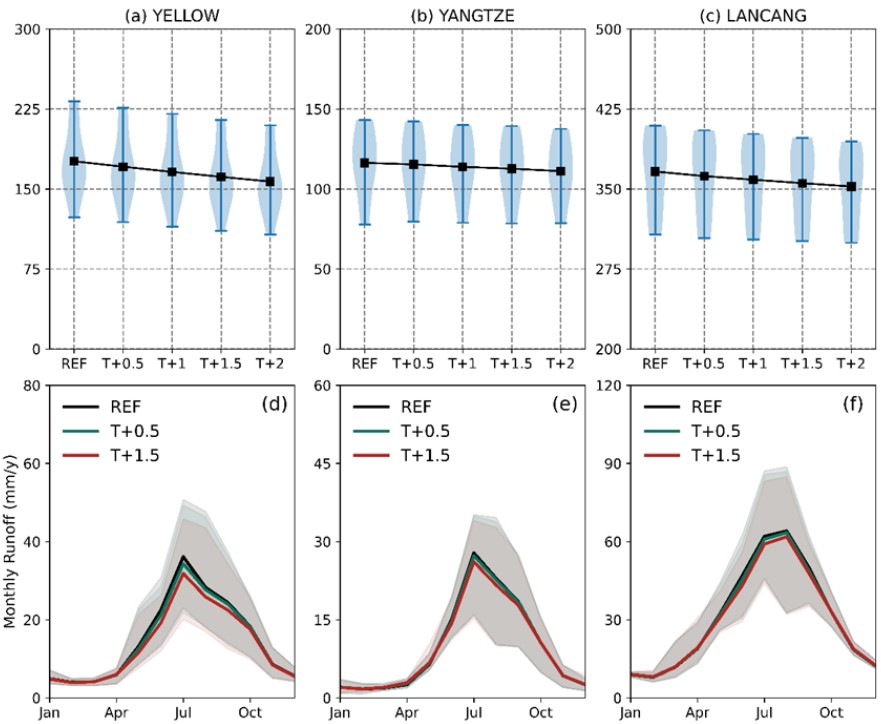

**Figure 8.** Same with Figure 6 but for the runoff response to the perturbed temperature.

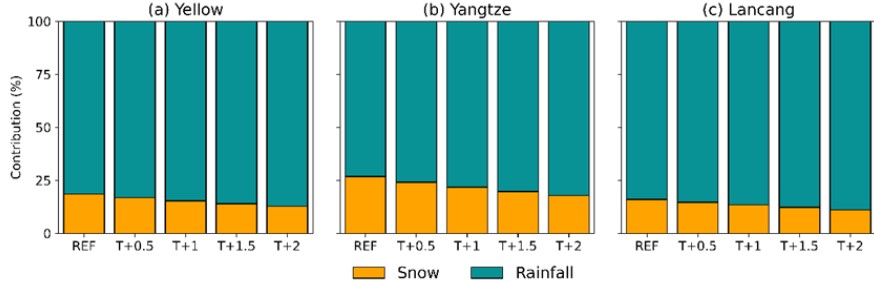

**Figure 9.** The runoff components with different perturbed temperature scenarios in Yellow, Yangtze, and Lancang, respectively.

increases. With a 2 °C temperature rise, the contribution of snowfall runoff decreases by 5.8%, 8.9%, and 5.0% in the Yellow,
Yangtze, and Lancang basins, respectively. These results strongly support the credibility of our analysis.

## 5   Conclusions

In this study, we propose hybrid distributed hydrological models that synergize the distributed process-based model with
embedded neural networks (ENNs). The hybrid models use the distributed process-based model as the backbone, with ENNs





parameterizing and replacing internal modules. Taking three large alpine basins on the Tibetan Plateau as the study basins, the proposed models are test and compared with state-of-the-art models. The climate perturbation method is further carried out to test the applicability of the hybrid models to analyze the hydrological sensitivities to climate change in large alpine basins. Our main findings are as follows:

1. The optimal hybrid distributed model achieves superior performance in runoff modeling, with $NSE$ of higher than 0.87, approaching the state-of-the-art DL models and outperforming traditional process-based models. The optimal hybrid distributed model also demonstrates remarkable prowess in hydrological modeling at ungauged sites within the basin.

2. Further experiments reveal that the inclusions of ENNs for parameterizing and replacing modules can lead to higher model accuracy. Considering spatial information within the basin and introducing temperature in $ENN_Q$ to represent the soil freeze-thaw process also show enhanced predictive capabilities in hybrid models.

3. The results about hydrological sensitivities to climate change show resonable patterns: runoff exhibits an amplification effect in response to precipitation changes, with a 10% precipitation change resulting in a 15–20% runoff change in large alpine basins. Annual runoff exhibits greater sensitivity to increases in precipitation compared to decreases. The increase in temperature enhances evaporation capacity and reduces the contributions of snowfall runoff, leading to a decrease in the total runoff and a reduction in the intra-annual variability of runoff. With a 2 °C temperature rise, the contribution of snowfall runoff decreases by 5.8%, 8.9%, and 5.0% in the Yellow, Yangtze, and Lancang basins, respectively.

In summary, we provide an effective and easily interpretable hybrid distributed hydrological model and enhance our understanding about hydrological sensitivities to climate change in large alpine basins. However, this study is limited to only using three large alpine basins on the Tibetan Plateau to evaluate proposed hybrid models due to the limitation of computational resources. Future research will focus on extending the evaluation of the hybrid distributed model to encompass a broader range of basins.

*Code and data availability.* DEM (http://www.gscloud.cn/sources/details/310?pid=302), LAI (https://doi.org/10.5067/MODIS/MOD15A2H.006), CMFD (https://doi.org/10.11888/AtmosphericPhysics.tpe.249369.file), NDVI (https://doi.org/10.5067/MODIS/MOD13A3.0 06), HWSD (https://data.tpdc.ac.cn/zh-hans/data/3519536a-d1e7-4ba1-8481-6a0b56637baf/?g=HWSD). The observed runoff data and the threw model code are not publicly available due to the privacy reasons. The hybrid models code will be open once the manuscript is accepted.

## Appendix A

### A1 Distributed EXP-Hydro model equations

The distributed EXP-Hydro model firstly delineates the basin into many sub-basins. In each sub-basin, the lumped EXP-Hydro is run independently (Equation A1-A12) to obtain the respective runoff. The runoff from all sub-basins is then aggregated to calculate the basin runoff (Equation A12). The detailed equations are as follows (Patil et al., 2014).





(1) Water balance

$$\frac{dS_0}{dt} = P_s - M \tag{A1}$$

$$\frac{dS_1}{dt} = P_r + M - ET - Q \tag{A2}$$

where $S_0$, $S_1$, $P_s$, $P_r$, $M$, $ET$, and $Q$ are snow storage, basin water storage, snowfall, rainfall, snowmelt, evaporation, and runoff, respectively.

(2) Precipitation partition

$$P_s = \begin{cases} 0 & T > T_{min} \\ P & T \leq T_{min} \end{cases} \tag{A3}$$

$$P_r = \begin{cases} P & T > T_{min} \\ 0 & T \leq T_{min} \end{cases} \tag{A4}$$

where P and T are precipitation and air temperature.

(3) Snowmelt

$$M = \begin{cases} min\{S_0,\ D_f \cdot (T - T_{max})\} & T > T_{max} \\ 0 & T \leq T_{max} \end{cases} \tag{A5}$$

(4) Evapotranspiration

$$ET = \begin{cases} 0 & S_1 < 0 \\ PET \cdot \left(\frac{S_1}{S_{max}}\right) & 0 \leq S_1 \leq S_{max} \\ PET & S_1 > S_{max} \end{cases} \tag{A6}$$

$$PET = 29.8\, L_{day} \frac{e_{sat}(T)}{T + 237.3} \tag{A7}$$

$$e_{sat}(T) = 0.611 \times \exp\left(\frac{17.3T}{T + 237.3}\right) \tag{A8}$$

where $PET$, $L_{day}$, and $e_{sat}(T)$ represent the potential evaporation, day length, and the saturation vapor pressure.

(5) Runoff and baseflow

$$Q_b = \begin{cases} 0 & S_1 < 0 \\ Q_{max} \cdot e^{-f \cdot (S_{max} - S_1)} & 0 \leq S_1 \leq S_{max} \\ Q_{max} & S_1 > S_{max} \end{cases} \tag{A9}$$





$$Q_s = \begin{cases} 0 & S_1 \leq S_{max} \\ S_1 - S_{max} & S_1 > S_{max} \end{cases} \tag{A10}$$

$$Q = Q_b + Q_s \tag{A11}$$

where $Q_b$ and $Q_s$ are the baseflow generated depending on the available storage in the basin bucket and the capacity-excess
runoff generated when basin bucket is saturated. All above undefined variables are calibration parameters. The details please
refer to (Patil and Stieglitz 2014).

(6) Basin runoff

$$Q_{basin} = \frac{\sum_{i=1}^{N} Q_i * A_i}{\sum_{i=1}^{N} A_i} \tag{A12}$$

where $Q_{basin}$ is the runoff at basin outlet. $Q_i$ and $A_i$ are the runoff and area of sub-basin $i$. $N$ is the total number of sub-basins
within the basin.

**A2 Hybrid distributed model equations**

In all hybrid distributed models, four ENNs are constructed to parameterize ($NN_\theta$) and replace runoff ($NN_Q$), precipitation
partition ($NN_S$), and snowmelt processes ($NN_M$). The detailed equations are as follows.

$$\theta_d = NN_\theta (A_s) \tag{A13}$$

$$Q = NN_Q (M + P_r, S_1, T, A_s) \tag{A14}$$

$$P_s = P \times NN_S (P, T, A_s) \tag{A15}$$

$$P_r = P - P_s \tag{A16}$$

$$M = S_0 \times NN_M (T, A_s) \tag{A17}$$

where $\theta_d$ and $A_s$ represent calibration parameters and static basin attributes, respectively. Detailed static basin attributes
refer to Table A1.



**Table A1.** The summary of static basin attributes for the inputs of ENNs.

| Variables | Descriptions | Units |
|---|---|---|
| P_mean | Mean daily precipitation | mm/d |
| T_mean | Mean daily air temperature | mm/d |
| PET_mean | Mean daily potential evaporation | mm/d |
| Basin area | Basin area | km2 |
| SLOPE_mean | Mean slope | m/km |
| DEM_mean | Mean elevation | m |
| Aridity | PET/P | - |
| LAI_max | Maximum monthly of the LAI | - |
| LAI_diff | Difference between maximum and minimum monthly mean of the LAI | - |
| River length | The river length from a sub-basin to the basin outlet | km |

*Author contributions.* BL conceived the idea and collected the data. BL, TS, FT, and GN conducted the analysis. BL drafted the manuscript and all authors reviewed and edited the manuscript.

*Competing interests.* At least one of the (co-)authors is a member of the editorial board of the Hydrology and Earth System Sciences.

*Acknowledgements.* This work was funded by Gansu Province Science and Technology Department (22ZD6WA043) and National Natural
Science Foundation of China (92047301).



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
