# Peer review of "Hybrid Hydrological Modeling for Large Alpine Basins: A Semi-distributed Approach"

_Hydrology and Earth System Sciences, 2024_

## Author Comment (AC3)

**1. Scope**

The scope of the paper is well suited for HESS.

**2. Summary**

The authors based their study on the hybrid model proposed by Li et al. (2023b), in which different modules of a conceptual hydrological model are substituted by Neural Networks. The authors then proposed a distributed version of the model, in which they subdivide the basins of interest into smaller subbasins and apply the hybrid models to each of these subbasins.

The authors then compare the performance of the lumped and distributed hybrid models against purely data-driven techniques (LSTM and CNN-LSTM) and show all models achieved similar performance. They also test the performance of the hybrid models to predict discharges in some of the predefined subbasins (untrained gauges). In the last sections, the authors run some experiments looking at the behaviour of the models when boundary conditions are modified (changes in precipitation and temperature).

**3. Evaluation**

Overall, the manuscript has the potential to be a good contribution, however, there are certain aspects mentioned in the questions below that should be taken into account before moving on to the next steps.

_**Response:**_ Thanks for your recognition and valuable suggestions. Please find our replies below.

**3.1 Major comments:**

- **The code is not published, and the authors indicate that it will be opened once the manuscript is accepted. I strongly recommend the editor to ask for an open code during the review process, as it increases the transparency of the study. I also tried to look for the code of the previous study (Li et al., 2023b) however I was not able to find it.**

  _**Response:**_ Thanks for your suggestions. We will open the source code in the revised manuscript.

- **They also indicate that the discharge information is not publicly available due**

**to privacy reasons. Even though this reason is valid and outside of the capabilities of the authors, it automatically makes the study non-reproducible, which is especially important when machine learning methods are being proposed.**

*Response:* Thanks for your suggestions. To enhance the transparency and reproducible of our study, we will provide the simulated results in Yellow in the revised manuscript.

- **The printing quality of all figures should be improved. When I zoom in, I cannot see the details. I suggest the authors print the figures in 300 dpi.**

*Response:* We will improve the printing quality in the revised manuscript.

- **The authors do not show the subbasins they used to create the distributed model. I encourage them to include this information. Also, I was not able to find information of the amount of subbasins they used.**

*Response:* We used the green lines in Figure 2 to show the delineated river networks within three basins, which determines the shape and number of delineated sub-basins. Referring to the number of subbasins divided by THREW model, we delineated the Yellow, Yangtze, and Lancang into 83, 99, and 63 subbasins. The detailed subbasins information will be added in the revised manuscript.

- **One major concern is that they are not considering any routing method. Consequently, even with a distributed model they just sum up the discharge coming from each subbasin. Moreover, the authors are working with large basins (over 90 000 km2 according to the manuscript) in which the routing processes can become highly relevant. Is there a reason why no routing is being used?**

*Response:* We agree with the reviewer that the routing method is important for hydrological modeling, especially in large basins. In this study, to achieve the coupling between physical models and neural networks and the simultaneous training of both the physical models and neural networks, all equations are fomulated to be differentiable to ensure operating within the differential programming framework (DPF). The technical requirements of DPF limit the consideration of routing methods in our hybrid hydrological models. To compensate for the lack of consideration of the routing process,

we calculate the river length from each sub-basin to the basin outlet and employ this static attribute as the inputs of ENNs to implictly characterize the routing process within the basin. We will discuss this limitation in the revised manuscript.

■ **The good performance of the distributed models can also be attributed to the fact that one has a more flexible model. More flexible models can get a better fit to the data, but this is not directly related to having a distributed version. One way to test this hypothesis is to use the same number of models you used in the distributed version but let them receive the same data (similar to Feng, 2022). If your models performed better, then you can say that the distributed nature of the models is beneficial, and the improvement is not just because of the increase in flexibility. If the performance is the same, then it would mean that the distributed version (especially without any routing) does not give an advantage.**

_**Response:**_ Feng et al. (2022) conceptualized each basin as being composed of multiple parallel components to represent spatial heterogeneity. The inputs of all components are assumed to be same and the parameters in different components are acquired independently. So theses models only enhanced model flexibility by seting more parameters but did not represent the spatial heterogeneity in the real world. Different from Feng et al. (2022), we developed hybrid models in a distributed manner similar with traditional distributed hydrological models. All parameters of physical models are calculated based on the static attributes of sub-basins. And all ENNs for replacement of different internal modules utilized static attributes and dynamic driving variables as inputs.Our models can better represent spatial heterogeneity and are less flexiable than models of Feng et al. (2022). And if the model inputs of all sub-basins are set the same, parameters in all sub-basins will be same, which means that all outputs are also same and the final results are same or similar with lumped models. Besides, Patil and Stieglitz (2015) demonstrated that our backbone model, the distributed EXP-Hydro model, outperform the lumped version by capturing spatial heterogeneity. Therefore, compared with lumped models, our distributed hybrid models are enhanced due to the consideration of spatial heterogeneity but not the improvement of model flexibility.

- **In Section 3.2 the authors present the results of the evaluation on untrained gauges. One can see that for the MT subbasin the performance of three models completely drops when compared to the performance in the original basin (TNH). Why? In the text it was mentioned that "the models show comparatively poorer performance in runoff modeling at the MT station" but it was never explained why. Also, the discharge range (Figure 5) of the subbasins is similar, so is there a reason for the big differences in performance?**

*Response:* Thanks for your suggestion. The upper reaches of the source region of the Yellow (the basin above the MT Hydrological Station) are characterized by high altitude and low temperatures. The alpine hydrological processes, including soil freeze-thaw, snow, and ice processes, significantly impact runoff processes. These influences are more pronounced compared to the entire source region of the Yellow, making hydrological modeling more challenging. The distributer EXP-Hydro, lacks the capability to accurately depict these alpine hydrological processes, resulting in relatively low simulation accuracy. In contrast, the ENNs, which represent these processes, are better at capturing them, demonstrating superior accuracy. This also proves that the coupled model constructed in this study can effectively learn and represent these hydrological processes. We will add detailed explain in the revised manuscript.

**3.2 Minor comments:**

- **Line 25: Is the word 'almost' a typo?**

*Response:* "almost" has been deleted in the revised version.

- **Line 32: Saying that process-based models can be used to understand the entire hydrological system including all internal processes is an overstatement, especially if you are referring to conceptual models. Conceptual models are mostly based on parameterized (empirical) relationships that somehow account for our understanding of the system, however the physics behind them is not much.**

*Response:* We agree with the reviewer that it is an overstatement. This sentense is revised as "They can be used to advance scientific understanding about the hydrological

systems and provide the insight into the response of hydrological processes to climate changes".

- **Line 122: It is not clear what the authors mean. Are the parameters the same for all basins or do they change?**

*Response:* We want to show that the parameters of all sub-basins within the basin are assumed to be the same in distributed EXP-Hydro model. It has been revised as "The calibration parameters of all sub-basins within the basin are assumed to be the same in distributed EXP-Hydro model, while many of them related to sub-basin attributes should be different".

- **Line 145: It would be better to create a table to specify the characteristics of each model.**

*Response:* Thanks for your suggestion. We will create a table to specify the characteristics of each model.

**Table 1.** Design details of different hybrid models. "√" represents that the model employs the corresponding ENNs while "×" means not.

| Model | $ENN_\theta$ | $ENN_S$ | $ENN_M$ | $ENN_Q$ | Temperature is the input of $ENN_Q$ |
|---|---|---|---|---|---|
| *DM* | × | × | × | × | × |
| $DM_\theta$ | √ | × | × | × | × |
| $DM_{\theta-Q}$ | √ | × | × | √ | × |
| $DM_{\theta-Q-T}$ | √ | × | × | √ | √ |
| $DM_{\theta-QSM}$ | √ | √ | √ | √ | × |
| $DM_{\theta-QSM-T}$ | √ | √ | √ | √ | √ |

- **Line 158: Hochreiter and Schmidhuber, 1997 created the LSTM architecture, but in this line, it sounds like they proposed the architecture for hydrological modelling. This paper should of course be cited, but not mixed with the other papers of hydrological applications.**

*Response:* We have revised this citation in the next version.

- **Figure 2. How was the spatial discretization of the basins? This should be included in the paper.**

***Response:*** Similar with the fourth comment and reponse in the major comment, we will add the detailed description about spatial discretization of the basin.

- **Line 195: What does suites of experiments mean? Shouldn´t it be set of experiments?**

***Response:*** We will revised as "set of experiments" in the revised manuscript.

- **Line 242-244: The PFAB values show a difference, but the changes of 0.01 in NSE are not significant. This can be just because of the initialization of the model. I do not agree that there is evidence to support that one model has an augmented ability to simulate overall runoff process.**

***Response:*** We agree with the reviewer that the $DM_\theta$ model cannot be demonstrated to achieve an augmented ability to simulate overall runoff process. It has been revised as "Specifically, the comparion results show that $DM_\theta$ model exhibits a closed but slightly better performance than the $DM$ model in overall runoff modeling, with an increase $NSE$ and $mNSE$ of 0.01-0.03 in all three basins. Additionally, lower $PFAB$ results imply that the $DM_\theta$ model contributes to an improved performance in peak runoff modeling. The incorporation of ENNs to represent spatial heterogeneity of calibration parameters can reduce the peak simulation biases and slightly improve the overall performance."

- **Figure 3: The size of the figure should be increased, right now it is hard to see the details of the hydrographs. Also, the authors should use the same line width for all models. Right now, some lines look thicker than others, which gives a bias to the figure. The last hydrograph does not have values on the y-axis.**

***Response:*** We will increase the size of all figures and revise the line width in Figure 3 in the revised manuscript.

- **Line 263: The authors say that when one includes air temperature in the ENN there is an evident enhancement of the model performance. The PFAB values vary a bit more, but the differences in NSE values are extremely small (0.01 for 4 cases, 0.02 for 1 case and 0.03 for another). This is just one metric summarizing more than 5 years of data. I do not agree that NSE values show an evident enhancement of the model performance.**

***Response:*** We agree with the reviewer. This statement has been revised as "Results indicate that $DM_{\theta-QSM-T}$ and $DM_{\theta-Q-T}$ models exhibit improved performance in peaking runoff modeling compared to the $DM_{\theta-QSM}$ and $DM_{\theta-Q}$ models, respectively. This enhancement is evident through closed $NSE$ and $mNSE$ and lower $PFAB$ in all three basins".

- **Figure 5e: How are the attributes being normalized? I do not understand how the area of the subbasins is comparable with the area of the entire basin. Also, why is the figure showing a range when referring to static attributes? This figure is not explained in the text.**

***Response:*** Thanks for your comment. Figure 5e shows 9 static attributes, except river length in Table A1, of all sub-basins within the basin. These static attributes are additional inputs of ENNs to capture the spatial heterogeneity among sub-basins among basins. Four lines represent static attribute ranges of the four basins with the four hydrological stations (TNH, JG, MQ, and MT) as the outlet. Among them, the basin with the TNH as the outlet is the largest basins and contains other three basins. These attributes of all sub-basins in four basins are normalized based on Equation 1. The mormalized attributes are final inputs of ENNs. Figure 5 are intended to show the meteorological and hydrological difference among four basins. We will add some discription about Figure 5 in the revised manuscript.

$$x_N = \frac{x - x_{min-TNH}}{x_{max-TNH} - x_{min-TNH}} \tag{1}$$

$x$ and $x_N$ represent the initial and normalized attributes, respectively. $x_{min-TNH}$ and $x_{max-TNH}$ represent the minimum and maximum variables among sub-basins within the basin with the TNH as the outlet.

- **Figure 7. The figure title indicates that the grey and yellow shading indicate annual and monthly responses. However, there are no shadings in the figure.**

***Response:*** We want to illestrate that annual responses are showed in the grey background while monthly responses in the yellow background. The caption of Figure 7 is revised as "Relative change of annual (grey background) and monthly(yellow background) runoff response to the perturbed precipitation (a-c) and air temperature (df) in Yellow, Yangtze, and Lancang, respectively".

- **Figure 8. I suggest the authors use a proper name for the figure and not just refer to another figure.**

*Response:* It will be revised as "**Figure 8**. Runoff responses to altered temperature in the Yellow, Yangtze, and Lancang basins (a-c for annual; d-f for monthly). The error bars in panels a-c and the shaded areas in panels d-f denote the range of simulated runoff"

Feng, D., Liu, J., Lawson, K., et al. (2022). Differentiable, Learnable, Regionalized Process-Based Models With Multiphysical Outputs can Approach State-Of-The-Art Hydrologic Prediction Accuracy. Water Resources Research 58(10).

Patil, S.D. and Stieglitz, M. (2015). Comparing spatial and temporal transferability of hydrological model parameters. Journal of Hydrology 525, 409-417.

---

## Author Comment (AC5)

The paper developed a hybrid framework that integrates a distributed process-based hydrological model and embedded neural networks (ENNs) for streamflow modeling in large alpine basins. The distributed EXP-Hydro model uses multiple mathematical equations to describe hydrological systems, including precipitation, snowmelt, runoff, and baseflow, which can be replaced by neural networks. The hybrid framework performs well in both gauged and ungauged basins across three large alpine basins. My major concerns are as follows:

***Response:*** Thanks for your recognition and valuable suggestions. Please find our replies below.

**Major comments:**

1. I suggest the authors rewrite the abstract, as it is too long. Some sentences should be moved to the introduction or results sections of the manuscript.

***Response:*** Thanks for your suggestion. We will rewrite the abstract in the revised manuscript.

"Alpine basins are important water sources for human life and reliable hydrological modeling can enhance the water resource management in alpine basins. Recently, hybrid hydrological models, coupling process-based models and deep learning, exhibit considerable promise in hydrological simulations. However, a notable limitation of existing hybrid models lies in their failure to incorporate spatial information within the basin and describe alpine hydrological processes, which restricts their applicability in hydrological modeling in large alpine basins. To address this issue, we develop a set of hybrid distributed hydrological models by employing a distributed process-based model as the backbone, and utilizing embedded neural networks (ENNs) to parameterize and replace different internal modules. The proposed models are tested on three large alpine basins on the Tibetan Plateau. A climate perturbation method is further used to test the applicability of the hybrid models to analyze the hydrological sensitivities to climate change in large alpine basins. Results indicate that proposed hybrid hydrological models can perform well in predicting runoff processes and simulating runoff component contributions in large alpine basins. The optimal hybrid model with Nash-Sutcliffe efficiency coefficients (*NSEs*) higher than 0.87 shows

comparable performance to state-of-the-art DL models. The hybrid distributed model also exhibits remarkable capability in simulating hydrological processes at ungauged sites within the basin, markedly surpassing traditional distributed models. Besides, the results also show reasonable patterns in the analysis of the hydrological sensitivities to climate change. Overall, this study provides a high-performance tool enriched with explicit hydrological knowledge for hydrological prediction and improves our understanding about the hydrological sensitivities to climate change in large alpine basins."

2.  The differences between the distributed models and the corresponding lumped models are unclear. From the manuscript, it appears that the only difference is that the lumped model simulates discharge for the entire basin, while the distributed model simulates discharge for each subbasin, and then summarizes the discharge for all the subbasins. Runoff routing is an important process in distributed hydrological models, which is also crucial for large basins. Please explain why river routing is missing.

***Response:*** Thanks for your suggestion. In this study, we employ the distributed EXP-Hydro model as the backbone model. Compared with the lumped version, the distributed EXP-Hydro model first delineate the entire basin into many sub-basins, and all hydrological processes are calculated in each sub-basin. The final basin runoff is acquired by summing the runoff outputs from all basins. Besides, our hybrid models utilized ENNs to parameterize and replace internal modules. We used static basin variables as the inputs of ENNs to represent the spatial heterogeneity within different sub-basins. On the other hand, we agree with the reviewer that the routing method is important for hydrological modeling, especially in large basins. However, to achieve the coupling between physical models and neural networks and the simultaneous training of both the physical models and neural networks, all equations are formulated to be differentiable to ensure operating within the differential programming framework (DPF). The technical requirements of DPF limit the consideration of routing methods in our hybrid hydrological models. To compensate for the lack of consideration of the routing process, we calculate the river length from each sub-basin to the basin outlet

and employ this static attribute as the inputs of ENNs to implicitly characterize the routing process within the basin. We will discuss this limitation in the revised manuscript.

3. Please demonstrate the importance of using subbasins in alpine basins due to the significant variability of precipitation and temperature in space. Additionally, the sensitivity of the area threshold for the subbasins is not discussed in the manuscript. While the authors may have experience defining the threshold in Tibetan basins, it is unclear how this applies to other basins

*Response:* Thanks for your suggestion. Many studies have demonstrated that our study basins exhibit significant spatial heterogeneity in precipitation and air temperature due to large topographical variations and complex weather systems (Ma et al. 2018, You et al. 2015). We will add this discussion in the revised manuscript. Besides, we used the green lines in Figure 2 to show the delineated river networks within three basins, which determines the shape and number of delineated sub-basins. Referring to the number of sub-basins divided by the THREW model, we delineated the Yellow, Yangtze, and Lancang into 83, 99, and 63 sub-basins. The detailed sub-basins information will be added in the revised manuscript.

4. The significance of model performance is not discussed in the manuscript. For example, DMθ-Q-T and DMθ-QSM-T have very close NSE values in the Yellow River and Lancang River. If the authors only trained the model once, it is unclear if the differences are statistically significant.

*Response:* We agree with the reviewer that a slight improvement in the NSE does not significantly demonstrate an enhancement of the model. In the revised manuscript, we will reassess the improvements of these models to enhance the credibility of the results.

5. The authors conducted a series of sensitivity tests of runoff to climate change. However, it is difficult to explain the internal structure of a neural network and how we can trust the extrapolated results. For example, the model was not trained on a 20% increase in precipitation, meaning the perturbed scenarios are extrapolations. It would be more accurate to refer to this as model sensitivity to dynamic inputs rather than concluding runoff sensitivities to climate change.

***Response:*** Many studies demonstrated that the performance of deep learning in simulating data outside the training range is significantly lower than within the training range. In this study, we introduced certain physical mechanisms into the deep learning model to enhance the physical consistency of the simulation results. To evaluate the model's performance in simulating data outside the training range, we used the climate perturbation method to assess the sensitivity of runoff processes to changes in temperature and precipitation. Although we did not use the perturbed data for training, our results were compared with existing studies, demonstrating the reasonableness of our simulation results and the ability to analyze the sensitivity of runoff processes to climate change. Besides, numerous studies have employed similar methods, using physical hydrological models to evaluate the sensitivity of runoff processes to climate change (Cui et al. 2023). We will include additional explanations in the revised manuscript.

6. The improvement in streamflow estimation is important. However, it would be interesting to investigate when and where these improvements occur. Please analyze the spatial differences between the deep learning models and the EXP-Hydro model in simulated discharge

***Response:*** Thanks for your suggestion. This study employed three metrics, including NSE, mNSE and PFAB, to evaluate the model improvement in different aspects. To further investigate when and where these improvements occur, we will add some analysis in the revised manuscript.

7. I found it hard to follow many sentences; please polish the language. Some examples are listed below.

***Response:*** Thanks for your suggestion. We will polish the language in the full manuscript.

**Minor comments:**

1. Line 25: Alpine basins are important water sources, playing a crucial role in various aspects of human life and the environment, such as domestic water supply, irrigation, hydropower generation, and climate regulation. Please rewrite the sentence.

*Response:* Thanks for your suggestion. This sentence will be revised in the manuscript.

2. Line 26: The performance of a hydrological model can be accurate, to describe the model, use reliable could be better.

*Response:* The "accurate" has been revised as "reliable" in the revised manuscript.

3. Line 27: shorten the sentence and use 'climate change and adaption'.

*Response:* This sentence is revised as "Developing reliable hydrological models is crucial for managing floods and improving water use efficiency under climate change.".

4. Line 31: These models depend on physical laws and empirical knowledge.

*Response:* This sentence is revised as "These models depend on physical laws and empirical knowledge to describe physical processes and are grounded in well-defined physical mechanisms."

5. Line 32-34: The sentence is too long. In addition, are these hydrological models sufficient to understand all hydrological processes?

*Response:* It will be revised as "They can be used to advance scientific understanding about the hydrological systems and provide the insight into the response of hydrological processes to climate changes"

6. Line 41: streamflow/discharge forecasting, snow water equivalent modeling, and groundwater level mapping. Please rewrite the sentence.

*Response:* We agree with the reviewer and the rewritten sentence is "They showcased exceptional model performance across diverse hydrological domains, including streamflow/discharge forecasting (Kratzert et al. 2018, Lees et al. 2021, Liu et al. 2021), snow water equivalent modeling (Duan and Ullrich 2021), and groundwater level mapping (Nourani et al. 2022, Solgi et al. 2021). "

7. Figure 2. Please add some subplots to show the spatial variability of precipitation and temperature, which is the main reason for using the distributed schemes. Please show the subbasins and indicate the amount of subbasins.

*Response:* We agree with the revised manuscript and we will add some subplots to show the spatial variability of precipitation and air temperature and sub-basins.

8. Line 86: …the proposed models…

*Response:* Thanks for your suggestion and we will revise in the revised version.

9. Line 87-88: Can the ENNs produce optimal parameters?

***Response:*** The differential programming framework ensures that the training parameters of hybrid models are similar to those of the deep learning model. By utilizing sufficient observed runoff data, although it cannot ensure obtaining the optimal parameters, it does ensure that the parameters are as fully trained as possible.

10. Line 203: The training period is 26 years and the evaluation/testing period is only 6 years. Is this setting reasonable? Why not set the same length for the training and testing? Please explain.

***Response:*** The proposed hybrid models, similar to deep learning, have numerous parameters that need to be trained, requiring a large amount of observational data. Due to the limited availability of observed data, we set the training period to 26 years and the testing period to 6 years. To ensure a fair comparison, we set the calibration/training and validation periods for the comparison models, including the physical model and the deep learning model, to be the same as those for the hybrid models.

11. Line 247: I don't think an improvement of NSE from 0.06 to 0.09 is a substantial improvement. Please rewrite the sentence.

***Response:*** We agree with the reviewer and the "substantial" and "noteworthy" have been revised as "slight" and "small".

Cui, T., Li, Y., Yang, L., et al. (2023). Non-monotonic changes in Asian Water Towers' streamflow at increasing warming levels. Nat Commun 14(1), 1176.

Duan, S. and Ullrich, P. (2021). A comprehensive investigation of machine learning models for estimating daily snow water equivalent over the Western US. Earth and Space Science Open Archive.

Kratzert, F., Klotz, D., Brenner, C., et al. (2018). Rainfall–runoff modelling using Long Short-Term Memory (LSTM) networks. Hydrology and Earth System Sciences 22(11), 6005-6022.

Lees, T., Buechel, M., Anderson, B., et al. (2021). Benchmarking data-driven rainfall–runoff models in Great Britain: a comparison of long short-term memory (LSTM)-based models with four lumped conceptual models. Hydrology and Earth System Sciences 25(10), 5517-5534.

Liu, Y., Zhang, T., Kang, A., et al. (2021). Research on Runoff Simulations Using Deep-Learning

Methods. Sustainability 13(3), 1336.

Ma, Z., Xu, Y., Peng, J., et al. (2018). Spatial and temporal precipitation patterns characterized by TRMM TMPA over the Qinghai-Tibetan plateau and surroundings. International journal of remote sensing 39(12), 3891-3907.

Nourani, V., Khodkar, K. and Gebremichael, M. (2022). Uncertainty assessment of LSTM based groundwater level predictions. Hydrological Sciences Journal 67(5), 773-790.

Solgi, R., Loaiciga, H.A. and Kram, M. (2021). Long short-term memory neural network (LSTM-NN) for aquifer level time series forecasting using in-situ piezometric observations. Journal of Hydrology 601, 126800.

You, Q., Min, J., Zhang, W., et al. (2015). Comparison of multiple datasets with gridded precipitation observations over the Tibetan Plateau. Climate Dynamics 45, 791-806.

---

## Author Response (AR2)

Dear Editor and Reviewers:

We appreciate the constructive comments and suggestions which help improve the manuscript. We provide the point-by-point response to all the comments, each comment starting with "***Response***". We look forward to hearing from you at your earliest convenience.

**For Editor:**

**I agree with the comments of the reviewer that the model should be called semi-distributed rather than distributed. Please correct this in the title and through the manuscript, and address the other reviewer concerns.**

***Response***: Thanks for your recognition and suggestions. We have revised the manuscript according to all comments. Our models are called semi-distributed models in the revised manuscript.

**For Reviewer #1**

**The authors indicate they have a distributed model, but they are not using any routing method, which is especially important considering the size of the basins they are analyzing. In the response from the previous review, and in the revised manuscript they indicated that "The technical requirements of differential programming framework limit the consideration of routing methods in our hybrid hydrological models." I do see why this would be the case. In differentiable programming, you can include routing methods, and it has been done multiple times. Feng et al., (2022) (https://doi.org/10.1029/2022WR032404) included a routing routine and the end of the pipeline using a unit hydrograph, Bindas et al., (2024) (https://doi.org/10.1029/2023WR035337) did the routing using Muskingum-Cunge, and Yu et al., (2024) also showed a similar approach (https://doi.org/10.5194/hess-28-2107-2024). Differentiable programming is a flexible approach that allows for routing. Considering that the authors are acknowledging the limitation and indicating that: "Future research will focus on developing hybrid distributed including routing processes and extending the**

evaluation of the hybrid distributed model to encompass a broader range of basins." I would suggest that they rename their current approach to semi-distributed. It would be consistent in the sense that they are giving more flexibility to the model by considering subbasins, with the clear limitation that no routing is being done. I would also suggest indicating in the Abstract that it is semi-distributed because of the routing problem.

*Response*: Thanks for your suggestions. Our models are called semi-distributed models in the revised manuscript.

**Finally, about the argument that they indicated in the response to the previous review: "To compensate for the lack of consideration of the routing process, we calculate the river length from each sub-basin to the basin outlet and employ this static attribute as the inputs of ENNs to implicitly characterize the routing process within the basin" This is not fully correct. The river length of each subbasin could indeed give an idea to the model about the internal routing in each subbasin, but it does not create routing between subbasins, because it does not have information to do this.**

*Response*: Thanks for your suggestions. We used the river length from each sub-basin to the basin outlet rather than the river length of each sub-basin. Employing the former as the inputs of ENNs can represent the flow distance of runoff from the river channel of different sub-basins to the outlet of the total basin and further can implicitly characterize the routing process within the basin.

**Minor comments:**

**Line 137: Remove "in this study" from the end of the sentence, you already mention that in the beginning of the sentence.**

*Response*: Thanks for your suggestions. "in this study" has been removed in the revised manuscript.

**Line 247: In the previous review it was mentioned that the authors should be careful in reporting NSE differences of 0.01 and 0.02 as significant, because they can be associated with the stochastic optimization process. They corrected this, as**

**shown in line 240. However, in line 247 they also indicated that a difference between 0.06-0.09 is a small increase, which I think could be statistically significant, and therefore not a small increase.**

*Response*: We have revised it as your suggestion in the revised manuscript. The revised sentence is "First, the results between $DM_{\theta\text{-}Q\text{-}T}$ and $DM_\theta$ models show the significant improvement in runoff modeling brought by the incorporation of $ENN_Q$. This enhancement is illustrated by an increase in $NSE$ and $mNSE$ values, ranging from 0.06 to 0.09 in Yellow and Yangtze."

**Line 261: "This enhancement is evident through closed NSE and mNSE and lower PFAB values in all three basins." The first part is confusing. Why would an enhancement in performance be evident if you have similar NSE and mNSE metrics?**

*Response*: We want to illustrate that the model is enhanced in peaking runoff modeling. This sentence has been revised as "This enhancement in peaking runoff modeling is evident through closed $NSE$ and $mNSE$ and lower $PFAB$ values in all three basins."